# TEMINET: A Co-Informative and Trustworthy Multi-Omics Integration Network for Diagnostic Prediction

**DOI:** 10.3390/ijms25031655

**Published:** 2024-01-29

**Authors:** Haoran Luo, Hong Liang, Hongwei Liu, Zhoujie Fan, Yanhui Wei, Xiaohui Yao, Shan Cong

**Affiliations:** 1Qingdao Innovation and Development Center, Harbin Engineering University, Qingdao 266000, China; luohaoran@hrbeu.edu.cn (H.L.); zhoujie.fan@hrbeu.edu.cn (Z.F.); 2College of Intelligent Systems Science and Engineering, Harbin Engineering University, Harbin 150001, China; lh@hrbeu.edu.cn (H.L.); hongwei.liu@hrbeu.edu.cn (H.L.); wyhhit@163.com (Y.W.)

**Keywords:** multi-omics, graph-based neural network, trustworthy, disease diagnosis, biomarker identification

## Abstract

Advancing the domain of biomedical investigation, integrated multi-omics data have shown exceptional performance in elucidating complex human diseases. However, as the variety of omics information expands, precisely perceiving the informativeness of intra- and inter-omics becomes challenging due to the intricate interrelations, thus presenting significant challenges in the integration of multi-omics data. To address this, we introduce a novel multi-omics integration approach, referred to as TEMINET. This approach enhances diagnostic prediction by leveraging an intra-omics co-informative representation module and a trustworthy learning strategy used to address inter-omics fusion. Considering the multifactorial nature of complex diseases, TEMINET utilizes intra-omics features to construct disease-specific networks; then, it applies graph attention networks and a multi-level framework to capture more collective informativeness than pairwise relations. To perceive the contribution of co-informative representations within intra-omics, we designed a trustworthy learning strategy to identify the reliability of each omics in integration. To integrate inter-omics information, a combined-beliefs fusion approach is deployed to harmonize the trustworthy representations of different omics types effectively. Our experiments across four different diseases using mRNA, methylation, and miRNA data demonstrate that TEMINET achieves advanced performance and robustness in classification tasks.

## 1. Introduction

In light of recent advancements in the acquisition of high-throughput omics data, multi-omics integration is rapidly expanding as a research field with the aim of providing a more comprehensive understanding of the underlying biological processes and molecular mechanisms involved in complex diseases [1,2]. Compared to single-omics studies, integrating multiple types of omics data enables the capture of complementary information across various molecular layers, leading to a more holistic view of biological systems [3]. Traditional approaches often involve statistical tools, which may have a limited capacity to fully capture the complex, non-linear relationships present in multi-omics datasets. In recent years, the application of artificial neural networks (ANNs) to multi-omics studies has emerged as a promising avenue to address these limitations [4,5,6]. ANN approaches can learn intricate patterns within data, enabling more accurate predictions and classifications, as well as the identification of previously unexplored relationships between molecular entities. These methods have shown great potential in various biomedical applications, such as disease prediction, patient stratification, and the discovery of novel biomarkers [7].

Despite significant achievements, the stability and biological explainability of many existing multi-omics integration approaches remain underdeveloped, primarily because of the insufficient exploration of both intra- and inter-omics interaction information. The existing multi-omics integration strategies can be grouped mainly into three categories: early, intermediate, and late fusion [7,8,9]. Early integration, such as fully connected neural networks (FCNNs) [10,11,12] and autoencoders (AEs) [13,14,15], perform integration through concatenating representations without considering the complex inter-omics interactions. Intermediate fusion emphasizes the interactions within inter-omics, rather than perceiving each omics type individually. For example, variational AEs (VAEs) are widely used to perform a fusion of homogeneous data types in a joint manner [16,17]. Despite the advantages of coordinated representation learning in intermediate fusion, it operates under the presumption of an equal contribution from each modality. This presumption can be challenging for the fitting capabilities of ANNs, especially in cases of feature imbalance, severe missing modalities, or substantial noise interference. Considering the uncertainty of such situations, trustworthy learning has been adopted. It involves transforming the operation from feature embedding to a decision-level process, which ultimately stabilizes and enhances the outcomes of late integration strategies. Han et al. [18] introduced a dynamic fusion approach for multi-modal classification. This method employed true-class probability to assess the classification confidence across different omics and then performed adjustments through modality confidence weighting for integration. Wang et al. [19] proposed Mogonet, which employed a view correlation discovery network (VCDN) to integrate initial classifications instead of fusing features across modalities, thereby utilizing label-correlated information in the shared space to produce final classification labels. We observed that, by perceiving and integrating the informativeness of each modality and inter-modality from distinct samples in trustworthy multimodal learning, practical applications can be significantly enhanced. However, traditional intra-modality information embedding may inadequately capture the full scope of informativeness, primarily due to the biological explainability of intra-omics. Such studies may overlook crucial aspects of the underlying biological processes, potentially resulting in a limited comprehension of the intricate molecular networks that drive various biological systems [20]. Ultimately, this could result in challenges to effective integration.

In omics research, it is crucial to acknowledge the interconnections among molecular functions, given the multifactorial nature of complex diseases [21,22]. Instead of treating each genetic factor as an independent entity, this approach allows for a more comprehensive understanding. Consequently, more investigations are focused on constructing disease networks by reconfiguring omics data into graph-based structures, reflecting the growing recognition of the importance of contextualized molecular interactions in understanding disease mechanisms. For instance, Ramirez et al. [23] employed a graph convolutional network (GCN) approach for cancer classification, leveraging a framework of gene co-expression based on Spearman correlating analysis. Althubaiti et al. [24] utilized the DeepMOCCA framework, combining omics data with protein–protein interaction networks for improved cancer prognosis predictions. Tang et al. introduced SiGra [25] and SpaRx [26], which used graph-based approaches to decode complex spatial tissue structures and nuanced cellular drug responses, showcasing a deeper insight into the dynamics of molecular interactions in biomedical research. Furthermore, Xing et al. [27] adopted an approach involving a weighted correlating method to build prior knowledge graphs. This method was particularly advantageous for unraveling disease-specific complexities. Therefore, inspired by the effectiveness of network representations in omics studies, we were motivated to adopt a graph-based representation to enhance the informativeness of intra-omics. This approach is intuitive and easy to interpret. It has the potential to better preserve the inherent structure and capture functional interactions from omics data, ultimately leading to improved disease prediction performance.

Based on the above observations, we propose a multi-omics integration framework, named TEMINET, for disease-predictive diagnosis that leverages graph attention networks and an uncertainty-based trustworthy strategy. Specifically, we construct a disease-specific network for each omics data to represent large-scale, unstructured, and irregular data effectively. We apply hierarchical graph attention networks (GATs) to capture co-informative intra-omics representations. Then, a trustworthy learning mechanism is employed to assess the reliability of informativeness using uncertainty information. Combine beliefs fusion integrates informativeness and uncertainty into an inter-omics fusion embedding for subsequent tasks. We conduct extensive experiments to show the effectiveness and robustness of TEMINET for multi-omics prediction. Our results demonstrate that the combination of graph-based feature representation and uncertainty-based trustworthy learning integration surpasses the state-of-the-art models on four disease classification tasks. We further employ a global interpretation approach to identify important biomarkers and analyze their disease-related functional relevance.

## 2. Results

Our research evaluated the effectiveness of the proposed model relative to current methodologies across diverse classification tasks. We also explored the scalability through the incorporation of additional omics data types and conducted a robustness study to assess the generalizability of the model. In the comparative analysis of various methodologies, accuracy (ACC) is a common metric employed for binary and multi-class classifications. In addition to ACC, binary classifications also utilize the F1 score (F1) and the area under the receiver operating characteristic curve (AUC). For multi-class classifications, the ACC, the weighted average F1 score (F1_weighted), and the macro-averaged F1 score (F1_macro) are used. Our experimental framework replicated the same settings as those used in Mogonet [19] and five random experiments to report the mean and standard deviation of evaluation metrics. To demonstrate a significant improvement in our model compared to the suboptimal method, we conducted an independent *t*-test. An obtained *p*-value of less than 0.05 indicated the statistical significance of the improvement.

Fourteen computational approaches were investigated, including six classic classifiers employing early integration strategies: K-nearest neighbor classifier (KNN) [28], support vector machine (SVM) [29], Lasso [30], random forest (RF) [31], XGboost [32], and fully connected neural networks (NNs) [33]. Furthermore, five multi-omics classifiers were analyzed: group-regularized ridge regression [34], BPLSDA for projecting data into latent structures with discriminant analysis [35], block PLSDA with additional sparse constraints (BSPLSDA) [35], the concatenation of final representations (CF) [36] for late-stage multi-omics data, and gated multimodal fusion (GMU) [37] that integrates intermediate representations, along with three advanced methods, Mogonet [19], MODILM [38], and Dynamic [18].

### 2.1. Classification Performance Comparison

In Table 1, a comparative analysis between the proposed model and established methods on ROSMAP and LGG datasets is provided. Compared to alternative approaches, the outcomes demonstrated that our suggested model performed better in binary classification tests. TEMINET outperformed the suboptimal method in many evaluation metrics, though not in the AUC metric for the ROSMAP dataset. The outcomes demonstrated the advantages of integrating informative data utilizing a multi-level, graph-based attention framework and disease-specific networks. Moreover, the proposed model significantly exceeded the performance of MODILM (GAT). This advantage was likely due to the incorporation of uncertainty-based adaptive fusion, enhancing the capability of the model to select and utilize the informative modalities, thus ensuring a more precise characterization of each subject. As shown in Table 2, the proposed method continued to lead in overall performance, yet it exhibited slightly lower efficacy in the five-class BRCA task compared to the top-performing method in this specific area. This further indicated the strength of our model in leveraging disease-specific network analysis and graph attention mechanisms, underscoring its potential despite the room for improvement in certain multi-class classification tasks.

### 2.2. Ablation Comparison of the Model Key Component

In an extensive ablation study of our framework, as shown in Table 3, performance metrics were compared against established benchmarks. Our model incorporated an advanced uncertainty-based mechanism and omics-specific co-expression networks, and it achieved increases in ACC and F1, particularly within the ROSMAP dataset, with which it outstripped the GAT+NN model by 4.3% and 4.4%. However, there was a slight decrement in the AUC by 1.0% compared to the second-best models. This outcome suggested that, while our model advanced predictive accuracy, it did so with a trade-off in AUC performance, signaling an area for further refinement in balancing predictive precision with generalizability across diverse datasets. The result of the LGG and KIPAN datasets also revealed improvements across all metrics, reflecting the exceptional ability of our model to capture intricate data patterns. For the BRCA dataset, our model exhibited enhancements in both ACC and F1-M, yet it encountered a slight decrease in the F1-W score, indicating potential for further refinement in the equilibrium of precision and recall.

### 2.3. Ablation Study Comparing Integration Performance across Varied Omics Categories

In our investigation, we analyzed the effectiveness of different omics data combinations for classification performance. Figure 1 shows that using all three omics types outperformed combinations of just two. This result underlined the individual and substantial contributions provided via different omics categories. Moreover, it confirmed the advantage of integrating multiple omics approaches. Our results demonstrated that TEMINET significantly improved classification performance by integrating multi-omics informative data. Remarkably, these enhancements became more pronounced with the gradual inclusion of diverse omics types. This observation underscores the scalability and adaptability of our model, suggesting its potential for broader applications in the field of biomedicine.

### 2.4. Robustness Study Involving Comparisons with Advanced Methods

In the robustness experiments, we increased the masked ratio to heighten the uncertainty of a specified modality. This method assessed the robustness of our model by comparing the reduction in accuracy as the data became increasingly incomplete or uncertain. Figure 2 demonstrates that TEMINET exhibited superior stability, with consistently lower accuracy reduction ratios across all masked ratios, compared to Mogonet and Dynamic. The robustness of TEMINET was attributed to its use of a graph-based topology and an uncertainty-based trust mechanism. While Mogonet also employed a graph-based approach by constructing similarity graphs among samples and incorporating the VCDN trust mechanism, its performance was moderate. Conversely, the Dynamic method, which relied on an encoder network and a confidence-based trust mechanism, displayed a greater decrease in accuracy, indicating reduced robustness relative to TEMINET. The findings underscored that TEMINET not only maintained lower accuracy reduction ratios but also exhibited stability across various levels of data masking, highlighting the effectiveness of its graph-based approach and uncertainty-based trust mechanism in preserving model robustness. This robustness enhances the generalizability and applicability of the model.

### 2.5. Important Biomarkers Identified via TEMINET

In interpreting the model, the primary objective was to identify biomarkers of significance. As shown in Table 4, the five most discriminating biomarkers based on their differential values were reported. The biomarkers with identical values were assigned the same ranking. If the number of tied positions exceeded the threshold for reporting (i.e., five distinct ranks), a random set from the tied group was chosen to fulfill the report. Subsequently, we conducted a brief review of the existing literature to elucidate the biological significance and disease association of these identified biomarkers.

As shown in Table 4, existing advances in AD research have identified biomarkers associated with its pathogenesis. As the most significant mRNA biomarker of AD identified via TEMINET, *MEIS3* was revealed through differential expression analysis to be considerably linked with cognitive decline and increased neurofibrillary tangle density [39,40]. Complementing this, cg19485804 (*NGEF*) emerged from LASSO regression analysis as another insight [41], notably associated with the APP mutation in mouse models. The downregulation of *NGEF* in the CVN-AD model suggested a critical role in modifying actin dynamics and consequently disrupting neuronal growth cone motility [42]. Furthermore, the microRNA miR-132 has been associated with the progression of Aβ and tau pathologies, with its reduced levels in circulation suggesting its utility as a potential diagnostic insight for AD. In the realm of breast cancer, *CA9* has been identified as an mRNA biomarker. A study showed that BRCA patients with lower levels of *CA9* derive more benefit from adjuvant therapies, suggesting that *CA9* expression could be instrumental in tailoring patient-specific treatment plans [43]. The interaction between IGF2BP3, TRIM25, and miR-3614 delineated a novel regulatory pathway crucial for tumor cell proliferation. The protective role of IGF2BP3 in safeguarding TRIM25 mRNA from degradation and its influence on miR-3614 maturation presented new potential targets for therapeutic intervention in BRCA [44]. In LGG, our model did not reveal any methylation or miRNA biomarkers. However, *ADD3* was identified as the leading mRNA biomarker. Essential for actin cytoskeleton assembly, *ADD3* deficiency in GBM cells triggered pro-angiogenic signaling, enhancing VEGFR expression in endothelial cells, which could have implications for angiogenesis in LGG [45]. Suggested as a tumor suppressor and survival predictor on chromosome 10q, *ADD3* was valuable for prognostic assessments in LGG [46]. In kidney-related cancers, miR-126 has been recognized for its strong prognostic potential in clear-cell renal cell carcinoma (ccRCC) [47], while miR-1271, markedly upregulated in ccRCC tissues, has emerged as a significant marker for assessing disease severity [48]. These findings affirmed the robust capability of TEMINET in elucidating complex biological markers pertinent to disease mechanisms and therapy responsiveness.

## 3. Discussion

The advancement of high-throughput techniques and individualized healthcare approaches has produced various supervised data collections critical for predictive applications such as pinpointing disease conditions, classifying tumor stages, and distinguishing cancer subtypes. The fusion of multi-omics information has demonstrated enhanced efficacy in disease prediction compared to single omics approaches. For clinical applications, these integration models must not only provide precise diagnostic guidance but also cover a wide range of diseases. This underscores the necessity for models exhibiting high accuracy and strong generalization across diverse medical conditions.

To achieve optimal multi-omics integration, the informativeness of modality representation has increasingly attracted attention. On the one hand, this informativeness reflects the quality of features specific to each omics type. This quality is contingent not only on the methodologies employed for feature representation but also on the inherent quality of the data. This is because data quality can be compromised during collection, storage, and processing, leading to potential loss and degradation. On the other hand, the informativeness of a modality significantly determines its contribution to the integration process. This contribution is measured by the extent to which data from a modality can complement or enhance the understanding that other omics types provide. It concerns not merely the quantity of data each modality brings but also the unique biological insights it offers that cannot be captured by others. Therefore, evaluating the informativeness of each modality representation is essential to ensure that the most informative and relevant features are utilized for better predictive accuracy and a deeper biological understanding.

In this study, we introduced TEMINET, a framework optimized for the trustworthy integration of multi-omics datasets. The advanced performance of TEMINET can be attributed to the joint observation of intra-omics molecular interactions and inter-omics informativeness. The framework addressed multifaceted complexities in disease pathogenesis by amalgamating omics data with topological models. Our approach involved constructing individual graphs for each subject within all omics data. This data formation strategy allowed us to leverage the inner topological information among intra-omics molecules obtained from disease-specific data to improve model performance. It emphasized the importance of the interplay between genetic factors in revealing the underlying causes of diseases. Investigations have shown that TEMINET outperforms in various metrics across four distinct tasks, demonstrating that enhancing intra-omics informativeness can significantly improve the performance of a trustworthy learning strategy in multi-omics integration. The outperformance observed with four diseases, including one cerebral degeneration disease and three cancers, highlighted the generalizability and adaptability of TEMINET at the disease level. The robustness study also confirmed its generalizability, as a robust model produces more stable and reliable results, which are particularly essential in real-world application scenarios. Additionally, the combination ablation experiment conducted at the omics level confirmed the scalability of TEMINET, indicating that its integration capabilities significantly improve with the increasing variety of omics data types. Applied to four diverse diseases, TEMINET enhanced the understanding of disease mechanisms and patient stratification, revealing biomarkers as potential insights and offering precise classifications. These advancements assist healthcare professionals in developing personalized therapeutic interventions based on deeper insights into patient conditions.

However, the model exhibited several limitations. In comparison to other models, TEMINET demonstrated lower computational efficiency due to the construction of multiple omics-level networks for each sample, potentially posing challenges for practical deployment. Meanwhile, it also presented computational demands that became apparent when dealing with larger datasets, indicating potential scalability issues. While it concentrates on specific omics interactions, the model might overlook the broader dynamics between different omics. This oversight could reduce the AUC performance, indicating that a more balanced approach should be considered in future developments.

This study can be additionally extended towards some future directions. For example, one direction would be to extend the capability of TEMINET by incorporating spatial transcriptomics data [25]. This enhancement would enable the analysis of not only mRNA, methylation, and miRNA but also the exploration of the spatial dimensions of cellular behavior and interactions. Another direction would be to improve the computational efficiency of the model and make it applicable to a broader range of diseases, thereby enhancing its generalizability.

## 4. Materials and Methods

### 4.1. Overview of TEMINET

The proposed method is illustrated in Figure 3. It begins with constructing a co-expression graph for the omics data of each subject via weighted gene co-expression network analysis (WGCNA). The second step involves generating initial classification results for each omics data using a multi-level GAT process. This process utilizes three layers for the extraction of intra-omics features, encompassing $G_{0}$, $G_{1}$, and $G_{2}$. $G_{1}$ is updated from $G_{0}$, and $G_{2}$ is subsequently updated from $G_{1}$. Thirdly, the uncertainty of each initial distribution is parameterized using subjective logic. Based on the Dempster–Shafer theory, the integration of multi-omics evidence comes from the inference of overall uncertainty and classification probability. The whole inference process concludes with the final label prediction of each subject.

### 4.2. Dataset Overview

In our investigation, we performed analysis using four public benchmark datasets, including ROSMAP for binary classification (distinguishing between Alzheimer’s disease (AD) and a normal control (NC)), BRCA for the classification of breast-invasive carcinoma into PAM50 subtypes (including five categories), low-grade glioma (LGG) for distinguishing between grade II and grade III, and KIPAN, referring to the classification of renal cell carcinoma subtypes. All datasets were acquired from Wang et al. [19], and they each contained three types of omics data: mRNA expression, DNA methylation, and miRNA expression. Detailed information regarding data acquisition and preprocessing was available in [19]. In brief, features with no signal (mean=0) and those with low variances (standard deviation=0.1 for mRNA, 0.001 for DNA methylation, and 0 for miRNA) were filtered out. To optimize the feature selection, the ANOVA F-value and principal component analysis (PCA) were employed. Initially, ANOVA tests were utilized to preselect features, reducing the impact of redundant ones. Subsequently, PCA was applied to the preselected features, aiming for the first principal component to account for less than 50% of the variance, thereby avoiding the overrepresentation of any individual feature within the dataset. Each feature was then scaled to the range of [0,1] through a linear transformation. Comprehensive details regarding these datasets are presented in Table 5.

### 4.3. Intra-Omics Network Construction

The development of functional interaction networks is integral to understanding the pathogenesis of complex diseases. To leverage synergistic relationships among intra-omics molecules, the initial step in our methodology involved implementing the WGCNA [49]) to construct intra-omics co-expression networks, as shown in Figure 3B. The construction of the intra-omics graph $G_{0}$ for each subject involved several key stages: Firstly, an adjacency matrix was formed through the WGCNA, with each entry indicating the correlation strength between pairs of omics features. Secondly, this matrix was transformed into an edge matrix by applying a threshold. Thirdly, a co-expression network was constructed for individual subjects, assigning omics data expression values to nodes as their features.

In this context, an initial co-expression graph network of each patient was denoted as $G_{0}=G\left(V_{0}^{d\times 1},E_{0}^{d\times d}\right)$. Here, $V^{d\times 1}$ symbolizes the attributes of *d* nodes. $E^{d\times d}$ represents the edge matrix derived from the co-expression correlations computed via the WGCNA. Specifically, for each subject *n*, a feature vector of dimension 1×d was generated, where *d* represents the number of features. For a group of *N* subjects within a single omics data, an N×d matrix was formulated to calculate the co-expression matrix $A^{d\times d}$. The correlation $A_{ij}$ between node $v_{i}$ and $v_{j}$ was determined as follows:(1)$$A_{ij}={\left(\frac{1}{2}\left(1+\frac{\sum_{n=1}^{N}\left(v_{i,n}-{\bar{v}}_{i}\right)\left(v_{j,n}-{\bar{v}}_{j}\right)}{\sqrt{\sum_{n=1}^{N}{\left(v_{i,n}-{\bar{v}}_{i}\right)}^{2}}\sqrt{\sum_{n=1}^{N}{\left(v_{j,n}-{\bar{v}}_{j}\right)}^{2}}}\right)\right)}^{\beta },$$
where ${\bar{v}}_{i}$ and ${\bar{v}}_{j}$ denote the mean values of nodes $v_{i}$ and $v_{j}$, and β denotes an adjustable parameter set through WGCNA. The edge matrix $E_{0}^{d\times d}$ was then obtained by binarizing the values from the matrix $A^{d\times d}$. The optimal threshold for binarizing was determined through a grid search approach, where the threshold parameter was varied within a range of 0.05 to 0.5. The optimal threshold setting in our study was 0.08. Intra-omics co-expression networks for mRNA, methylation, and miRNA datasets were constructed similarly in our study.

### 4.4. Intra-Omics Informative Augmentation

To leverage the information embedding in nodes and their associations within co-expression matrices effectively, we introduced GAT [50] to enhance the disease-specific characteristics of the omics dataset. GAT incorporated masked self-attention-based layers to enable the dynamic weighting of neighboring node contributions, which allowed GAT to selectively focus on more pertinent adjacent nodes, thereby diminishing the impact of nodes that were less significant. As a result, GAT exhibited a superior ability to discern intricate and non-structure connections, as well as variations within the topology of the graph.

Specifically, an initial network for each omics subject *n*, denoted as $G_{0}=G\left(V_{0}^{n},E_{0}\right)$ in Figure 3B, was updated through a GAT layer. Initially, for a node $h_{i}$ within this network, the normalization of attention coefficients $\alpha_{ij}$ with its neighboring nodes $h_{j}$ was calculated as follows:(2)$$\alpha_{ij}=\frac{\exp(\mathrm{LeakyRelu}\left(\alpha \left(\left[W_{C}h_{i}\|W_{C}h_{j}\right]\right)\right))}{\sum_{h_{l}\in N_{i}}\exp\left(\mathrm{LeakyRelu}\left(\alpha \left(\left[W_{C}h_{i}\|W_{C}h_{l}\right]\right)\right)\right)},\;h_{j}\in N_{i},$$
where ‖ is the concatenation operation, and $W_{C}$ is the shared parameter matrix for linear transformation. To ensure a more stable self-attention update, a multi-head approach was introduced [50]. We conducted a process in which the attention-layer functions were implemented *T* times, each with unique parameters. The outcomes of these replications, indicated as ${h^{\prime }}_{i}$, then conducted an aggregating concatenation in sequence, as follows:(3)$${h^{\prime }}_{i}{=\|}_{t=1}^{T}\sigma \left(\sum_{j\in N_{i}}\alpha_{ij}^{t}W_{R}^{t}h_{j}\right),$$
where $\alpha_{ij}^{t}$ represents the attention coefficients from the *t*-th attention head, and $W_{R}^{t}$ is the weight matrix associated with the *t*-th head.

To enhance the exploration of internal feature relationships, we incorporated the multi-level feature representation approach implemented by Xing et al. [27]. The foundational network $G_{0}$ encapsulated data corresponding to the primal features. Subsequently, $G_{1}$ evolved from $G_{0}$ via a multi-head GAT attention layer. Analogously, $G_{2}$ was generated from $G_{1}$. This progression through three progressive graph network layers created a hierarchical integration structure, systematically amalgamating features across the GAT layers. Ultimately, the vectors produced from these transformation stages were fused, resulting in comprehensive feature representations. This design facilitates a dynamic optimization of feature interplays within the network, leading to a more substantial and comprehensive representation of the fundamental biology mechanisms. Using a similar process, we also implemented the DNA methylation and miRNA informative augmentation for each subject.

### 4.5. Trust-Driven Multi-Omics Fusion

In traditional multi-omics integration methods, the trustworthiness of different datasets is often not adequately considered, leading to potential inaccuracies in understanding complex biological processes. To address this, we introduced an uncertainty-based trustworthy learning approach to our integration method [51]. This approach enhances trustworthiness and precision in multi-omics data integration by quantifying inherent uncertainty in each modality. It leverages this measure of uncertainty to jointly perceive informativeness across intra- and inter-omics. Given that uncertainty assessments define confidence in predictions, the evidence of a dataset with lower uncertainty should achieve higher trustworthiness and be assigned a larger contribution to multi-omics integration.

Evidence in a classifier is generally considered the outputs of a neural network processed through an activation function like softmax. In our study, the evidence $e^{m}=\left[e_{1}^{m},\cdots ,e_{K}^{m}\right]$ for the $m^{th}$ omics category across *K* classes was generated from GAT-enhanced features. In the augmentation module, GAT-enhanced features were transformed into evidence through a sequence of fully connected layers and an active layer. Here, we set a cross-entropy loss $L_{GAT}^{m}$ to modify the GAT augmentation module.

Secondly, to obtain the uncertainty, we applied subjective logic [52] to the evidence $e^{m}=\left[e_{1}^{m},\cdots ,e_{K}^{m}\right]$. For each class *k* in the $m^{th}$ omics category, the belief mass $b_{k}^{m}$ and uncertainty mass $u^{m}$ were calculated:(4)$$u^{m}+\sum_{k=1}^{K}b_{k}^{m}=1,$$
where $u^{m}\geq 0$ indicates the overall uncertainty in the classification for the *m*th omics category and $b_{k}^{m}\geq 0$ indicates the confidence in each class prediction. The concentration parameters $\alpha^{m}$ of the Dirichlet distribution were determined from the evidence, where $\alpha_{k}^{m}=e_{k}^{m}+1$. The belief mass for each class was computed as $b_{k}^{m}=\frac{e_{k}^{m}}{S^{m}}$ and the overall uncertainty as $u^{m}=\frac{K}{S^{m}}$, with $S^{m}=\sum_{i=1}^{K}\left(e_{i}^{m}+1\right)=\sum_{i=1}^{K}\alpha_{i}^{m}$. An opinion, $M^{m}=\left[b_{1}^{m},b_{2}^{m},\cdots ,b_{K}^{m},u^{m}\right]$, was obtained for the evidence $e^{m}$. In summary, for the *m*th omics category, the more evidence gathered for each of the *K* classes, the higher the probability assigned to the respective class, thus reducing uncertainty. Conversely, a scarcity of evidence led to increased uncertainty. Utilizing subjective logic, this approach models second-order probabilities and uncertainties for the $m^{th}$ omics category, effectively countering the overconfidence often seen in traditional neural network classifiers.

Thirdly, within the methodological framework for multi-omics fusion, we applied the Dempster–Shafer theory to synthesize evidence from different classes. This process consolidates independent probability mass assignments from each class into a unified joint mass. The Dempster rule orchestrates this fusion to merge belief and uncertainty across the omics spectrum, symbolized as follows:(5)$$M^{F}=M^{1}⊕M^{2}⊕\cdot \cdot \cdot M^{m},$$
where $M^{F}$ denotes the combined beliefs, and $M^{m}$ represents the opinion of the $m^{th}$ omics type. As illustrated in Figure 3D, we took a fusion of two omics categories as an example. The first category, $M^{1}$, represented in orange, corresponded to the mRNA type and was denoted as $\left[b_{1}^{1},b_{2}^{1},\cdots ,b_{K}^{1},u^{1}\right]$. The second category, $M^{2}$, represented in green, corresponded to the methylation and was denoted as $\left[b_{1}^{2},b_{2}^{2},\cdots ,b_{K}^{2},u^{2}\right]$. In the process of combining these two sets, we focused on recombining the compatible elements (indicated with brown blocks) while disregarding the mutually exclusive parts (shown as white blocks). The fusion process was implemented to form the combined beliefs, which were denoted as $M^{F}$, as follows:(6)$$M^{F}=M^{1}⊕M^{2},$$
and
(7)$$b_{k}^{F}=\frac{1}{1-C}\left(b_{k}^{1}b_{k}^{2}+b_{k}^{1}u^{2}+b_{k}^{2}u^{1}\right),\;u^{F}=\frac{1}{1-C}u^{1}u^{2},$$
where $\frac{1}{1-C}$ denotes the scale factor for normalization. The term $C=\sum_{i\neq j}b_{i}^{1}b_{j}^{2}$ represents a degree of conflict observed between the two sets of mass values, and $b_{i}^{1}b_{j}^{2}$ represents the white blocks in Figure 3D. It can be observed that, in instances where both $M^{1}$ and $M^{2}$ exhibit high levels of uncertainty (with high values of $u^{1}$ and $u^{2}$), the resulting prediction manifests in low confidence, indicated by a lower value of $b_{k}$. Conversely, when both sources demonstrate low uncertainty (low values of $u^{1}$ and $u^{2}$), the resulting prediction exhibits high confidence, indicated by a higher value of $b_{k}$. When only one source shows low uncertainty (either $u^{1}$ or $u^{2}$), the final prediction relies on the more reliable source. After $M^{F}=\left\{{\left\{b_{k}^{F}\right\}}_{k=1}^{K},u^{F}\right\}$ was obtained, the final evidence could be inferred with $e_{k}=b_{k}\times S$, $\alpha_{k}=e_{k}+1$, and $S=\frac{K}{u}$.

Furthermore, we introduced an enhanced cross-entropy loss function by integrating sample-specific evidence, as follows:(8)$$L_{E-ce}\left(\alpha_{i}\right)=\sum_{k=1}^{K}y_{ik}\left(\psi \left(S_{i}\right)-\psi \left(\alpha_{ik}\right)\right),$$
where $\alpha_{i}$ is the parameter of the Dirichlet distribution for the *i*th sample, and ψ(·) is the digamma function. Building on this, an overall sample-specific loss function, which combined the adjusted cross-entropy loss with a Kullback–Leibler divergence term to effectively manage the evidence for incorrect labels, was defined as follows:(9)$$L\left(\alpha_{i}\right)=L_{E-ce}\left(\alpha_{i}\right)+\lambda_{t}KL\left[D\left(p_{i}|{\tilde{\alpha }}_{i}\right)\|D\left(p_{i}|1\right)\right].$$
The modified attribute of the Dirichlet distribution is ${\tilde{\alpha }}_{i}=y_{i}+\left(1-y_{i}\right)\alpha_{i}$, and λ is a balance factor greater than zero. This design helps penalize incorrect class evidence while preserving the evidence for the correct class. To ensure that the informative augmentation and evidence fusion were updated simultaneously, a total loss was defined as follows:(10)$$L_{global}=\sum_{i=1}^{N}\left[L\left(\alpha_{i}\right)+\sum_{m=1}^{M}L(\alpha_{i}^{m})+\gamma \sum_{m=1}^{M}L_{GAT}^{m}\right],$$
where γ denotes an adjusted attribute. We deployed γ=1 across our investigations.

### 4.6. Identifying Biomarkers with TEMINET

In the realm of biomedical research, the elucidation of biomarkers is pivotal for unraveling the intricacies of biological processes and providing insight into diverse outcomes. Concurrently, there is a growing need in clinical research for interpretable models that elucidate underlying disease mechanisms and bolster model credibility. Consequently, we applied a global interpretation method to identify the importance of biomarkers in our model. Specifically, to evaluate omics features in computational models, the normalization of these features on a scale from zero to one was initially implemented. Feature ablation involved individually removing features to evaluate their impacts on the model efficacy, with a focus on classification capability. The importance of each feature was determined by observing the reduction in model performance post-ablation. In binary classification and multi-class classification tasks, the F1 score and F1 macro score were used to assess the impact of feature ablation on model performance, respectively. This process was implemented with the best-performing model. For multi-omics data, we implemented this strategy with each type of omics data.

## Figures and Tables

**Figure 1 ijms-25-01655-f001:**
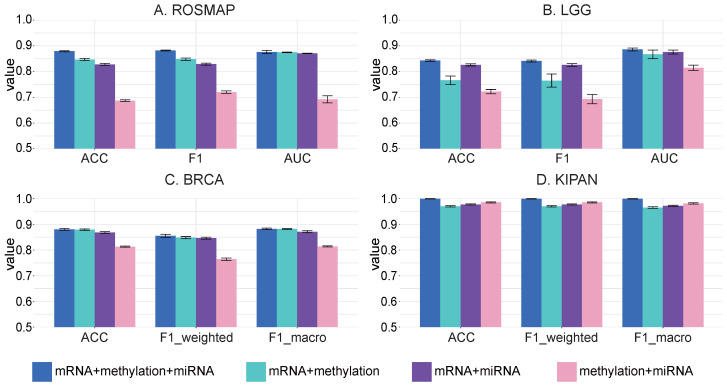
The efficacy of the proposed approach on different omics data combinations was assessed, presenting means and standard errors for comparison.

**Figure 2 ijms-25-01655-f002:**
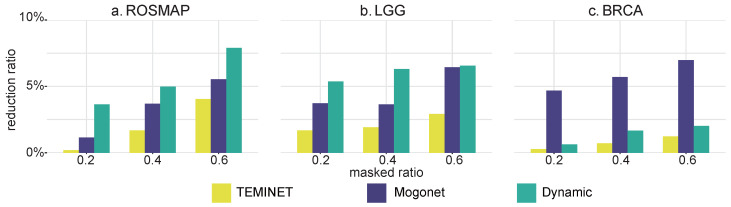
The robustness of the proposed approach was evaluated by comparing it with that of advanced methods. KIPAN was excluded from this comparison since it is relatively easy to classify.

**Figure 3 ijms-25-01655-f003:**
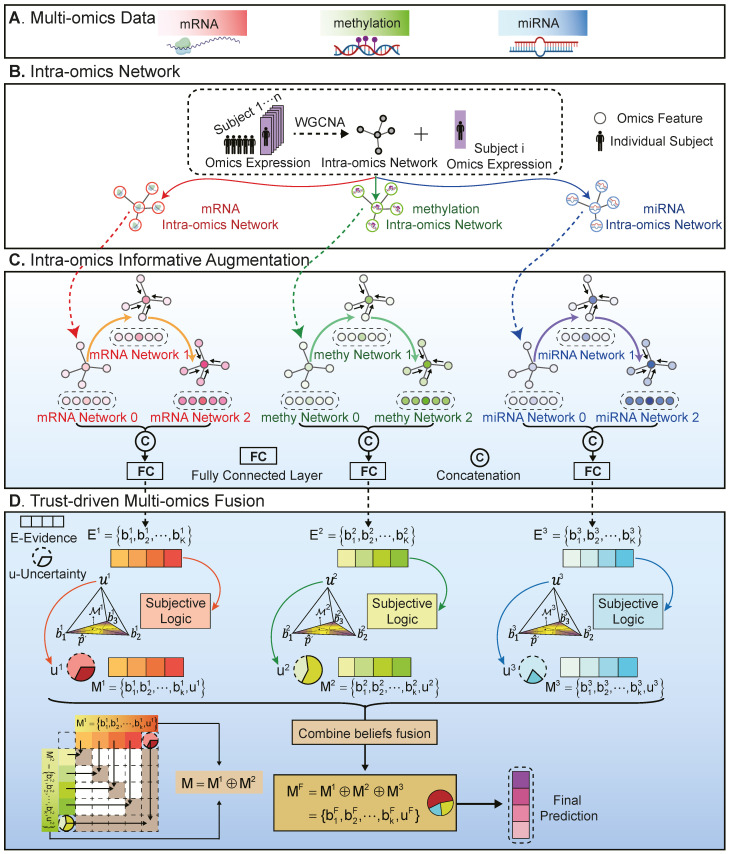
Framework of TEMINET. (**A**) TEMINET operates on a sample-wise basis with multi-omics information of each sample being imported into the model. (**B**) The first intra-omics network was built using the WGCNA. (**C**) The intra-omic information at each omics level was augmented using the multi-level GAT. (**D**) The evidence was evaluated via the subject logic module to determine the uncertainty. During the integration phase, the trustworthy informativeness and uncertainty of each omics were amalgamated into composite embedding that encompassed inter-omics information. The fusing representation was subsequently applied to implement a downstream classification task.

**Table 1 ijms-25-01655-t001:** Comparison with advanced methods using the ROSMAP and LGG datasets. The top-performing results are emphasized in boldface. Metrics marked with * indicate a significant improvement for our model compared to the suboptimal method, as confirmed with an independent *t*-test resulting in p< 0.05.

Method	ROSMAP (Two Categories)			LGG (Two Categories)		
	ACC (%)	F1 (%)	AUC (%)	ACC (%)	F1 (%)	AUC (%)
KNN	65.7 ± 3.6	67.1 ± 4.4	70.9 ± 4.5	72.9 ± 3.4	73.8 ± 3.3	79.9 ± 3.8
SVM	77.0 ± 2.4	77.8 ± 1.6	77.0 ± 2.6	75.4 ± 4.6	75.7 ± 5.0	75.4 ± 4.6
Lasso	69.4 ± 3.7	73.0 ± 3.3	77.0 ± 3.5	76.1 ± 1.8	76.7 ± 2.2	82.3 ± 2.7
RF	72.6 ± 2.9	73.4 ± 2.1	81.1 ± 1.9	74.8 ± 1.2	74.2 ± 1.0	82.3 ± 1.0
XGBoost	76.0 ± 4.6	77.2 ± 4.5	83.7 ± 3.0	75.6 ± 4.0	76.7 ± 3.2	84.0 ± 2.3
NN	75.5 ± 2.1	76.4 ± 2.1	82.7 ± 2.5	73.7 ± 2.3	74.8 ± 2.4	81.0 ± 3.7
GRridge	76.0 ± 3.4	76.9 ± 2.9	84.1 ± 2.3	74.6 ± 3.8	75.6 ± 3.6	82.6 ± 4.4
BPLSDA	74.2 ± 2.4	75.5 ± 2.3	83.0 ± 2.5	75.9 ± 2.5	73.8 ± 3.1	82.5 ± 2.3
BSPLSDA	75.3 ± 3.3	76.4 ± 3.5	83.8 ± 2.1	68.5 ± 2.7	66.2 ± 3.0	73.0 ± 2.6
CF	78.4 ± 1.1	78.8 ± 0.5	88.0 ± 0.5	81.1 ± 1.2	82.2 ± 0.4	88.1 ± 0.4
GMU	77.6 ± 2.5	78.4 ± 1.6	86.9 ± 1.6	80.3 ± 1.5	80.8 ± 1.2	88.6 ± 1.2
Mogonet	81.5 ± 2.3	82.1 ± 2.2	87.4 ± 1.2	81.6 ± 1.6	81.4 ± 1.4	84.0 ± 2.7
MODILM	84.3 ± 1.2	85.0 ± 0.8	89.1 ± 1.2	82.8 ± 0.7	82.5 ± 2.3	86.1 ± 1.5
Dynamic	84.2 ± 1.3	84.6 ± 0.7	**91.2 ± 0.7**	83.3 ± 1.0	83.7 ± 0.4	88.5 ± 0.4
Ours	**87.9 ± 0.4 ***	**88.2 ± 0.4 ***	87.6 ± 1.3	**84.3 ± 0.7 ***	**84.1 ± 0.9**	**88.6 ± 1.2**

**Table 2 ijms-25-01655-t002:** Comparison with advanced methods on the BRCA and KIPAN datasets. The top-performing results are emphasized in boldface. Metrics marked with ⁎ indicate a significant improvement in our model compared to the suboptimal method, as confirmed with an independent *t*-test resulting in p< 0.05.

Method	BRCA (Five Categories)			KIPAN (Three Categories)		
	ACC (%)	F1-W ^1^ (%)	F1-M ^1^ (%)	ACC (%)	F1-W ^1^ (%)	F1-M ^1^ (%)
KNN	74.2 ± 2.4	73.0 ± 2.3	68.2 ± 2.5	96.7 ± 1.1	96.7 ± 1.1	96.0 ± 1.4
SVM	72.9 ± 1.8	70.2 ± 1.5	64.0 ± 1.7	99.5 ± 0.3	99.5 ± 0.3	99.4 ± 0.4
Lasso	73.2 ± 1.2	69.8 ± 1.5	64.2 ± 2.6	97.4 ± 0.2	97.4 ± 0.2	97.2 ± 0.4
RF	75.4 ± 0.9	73.3 ± 1.0	64.9 ± 1.3	98.1 ± 0.6	98.1 ± 0.6	97.5 ± 1.1
XGBoost	78.1 ± 0.8	76.4 ± 1.0	70.1 ± 1.7	99.3 ± 0.8	99.3 ± 0.8	98.9 ± 1.4
NN	75.4 ± 2.8	74.0 ± 3.4	66.8 ± 4.7	99.1 ± 0.5	99.1 ± 0.5	99.1 ± 0.5
GRridge	74.5 ± 1.6	72.6 ± 1.9	65.6 ± 2.5	99.4 ± 0.4	99.4 ± 0.4	99.3 ± 0.4
BPLSDA	64.2 ± 0.9	53.4 ± 1.4	36.9 ± 1.7	93.3 ± 1.3	93.3 ± 1.3	91.9 ± 2.1
BSPLSDA	63.9 ± 0.8	52.2 ± 1.6	35.1 ± 2.2	91.9 ± 1.2	91.8 ± 1.3	89.5 ± 1.4
CF	81.5 ± 0.8	81.5 ± 0.9	77.1 ± 0.9	99.2 ± 0.5	99.2 ± 0.5	98.8 ± 0.9
GMU	80.0 ± 3.9	79.8 ± 5.8	74.6 ± 5.8	97.7 ± 1.6	97.6 ± 1.7	95.8 ± 3.2
Mogonet	82.9 ± 1.8	82.5 ± 1.6	77.4 ± 1.7	**99.9 ± 0.2**	**99.9 ± 0.2**	**99.9 ± 0.2**
MODILM	84.5 ± 0.9	84.0 ± 1.6	80.4 ± 1.2	99.2 ± 0.8	99.2 ± 0.8	99.2 ± 0.8
Dynamic	87.7 ± 0.3	**88.0 ± 0.5**	84.5 ± 0.5	**99.9 ± 0.2**	**99.9 ± 0.2**	**99.9 ± 0.3**
Ours	**88.0 ± 0.8**	85.5 ± 1.3	**88.2 ± 0.7 ***	**99.9 ± 0.2**	**99.9 ± 0.2**	**99.9 ± 0.2**

^1^ The terms F1-M and F1-W denote the F1 macro and weighted scores, respectively.

**Table 3 ijms-25-01655-t003:** This study examined key components of TEMINET with benchmark datasets. The top-performing results are highlighted.

Dataset	Metric	NN ^1^	NN ^1^ + Trust	GAT + NN ^1^	Our Model
ROSMAP	ACC (%)	76.6 ± 2.3	82.5 ± 0.9	84.3 ± 1.6	**87.9 ± 0.4**
	F1 (%)	77.7 ± 1.9	82.3 ± 0.6	84.5 ± 1.6	**88.2 ± 0.4**
	AUC (%)	81.9 ± 1.7	**88.5 ± 0.6**	**88.5 ± 0.6**	87.6 ± 1.3
LGG	ACC (%)	74.0 ± 3.9	81.9 ± 0.8	82.4 ± 0.5	**84.3 ± 0.7**
	F1 (%)	75.6 ± 3.6	81.5 ± 0.4	82.2 ± 1.3	**84.1 ± 0.9**
	AUC (%)	82.4 ± 3.6	87.1 ± 0.4	87.7 ± 1.3	**88.6 ± 1.2**
BRCA	ACC (%)	79.6 ± 1.2	84.2 ± 0.5	87.7 ± 0.4	**88.0 ± 0.8**
	F1-W ^2^ (%)	78.4 ± 1.4	84.4 ± 0.9	**88.2 ± 0.5**	85.5 ± 1.3
	F1-M ^2^ (%)	72.3 ± 1.8	80.6 ± 0.9	84.8 ± 1.7	**88.2 ± 0.7**
KIPAN	ACC (%)	98.8 ± 1.1	99.7 ± 0.3	98.9 ± 0.4	**99.9 ± 0.2**
	F1-W ^2^ (%)	98.8 ± 1.1	99.7 ± 0.3	98.9 ± 0.5	**99.9 ± 0.2**
	F1-M ^2^ (%)	98.1 ± 1.6	99.4 ± 0.5	98.7 ± 0.7	**99.9 ± 0.2**

^1^ NN refers to a neural network. ^2^ The terms F1-M and F1-W denote the F1 macro and weighted scores, respectively.

**Table 4 ijms-25-01655-t004:** Top five significant disease-specific biomarkers identified using TEMINET.

Dataset	Omics Data Type		
	mRNA Expression	DNA Methylation	miRNA Expression
ROSMAP	*MEIS3*, *NPNT*, *KIF5A*, *GPIHBP1*, *SAMD4A*, *CDK18*	cg04126866, cg08367223, cg27091787, cg19485804, cg24192663	hsa-miR-132, hsa-miR-129-5p, hsa-miR-146b-5p, hsa-miR-129-3p, hsa-miR-143
BRCA	*CA9*, *GPM6B*, *FAM171A1*, *RARA*, *KLHL29*	*A2LD1*, *A2ML1*, *ABAT*, *ABCA13*, *ABCC11*	hsa-mir-3614, hsa-mir-3677, hsa-mir-760, hsa-mir-937, hsa-mir-1269
LGG	*ADD3*, *AGXT2L1*, *AMOT*, *DAAM2*, *FAM189A2*	—	—
KIPAN	—	*LOC649395*	hsa-mir-126, hsa-mir-1270-1, hsa-mir-1270-2, hsa-mir-1271, hsa-mir-145

**Table 5 ijms-25-01655-t005:** Overview of datasets used in this investigation.

Dataset	Source	Subjects	Amount of Origin Data				Amount of Data for Study		
			mRNA	Methy	miRNA		mRNA	Methy	miRNA
ROSMAP	AMPAD	Normal Control: 169,	55,889	23,788	309		200	200	200
		Alzheimer’s Disease: 182							
BRCA	TCGA	Luminal A: 436, Luminal B: 147, HER2-enriched: 46, Normal-like: 115, Basal-like: 131	20,531	20,106	503		1000	1000	503
LGG	TCGA	Grade II: 246, Grade III: 264	20,531	20,114	548		2000	2000	548
KIPAN	TCGA	KICH: 66, KIRC: 318, KIRP: 274	20,531	20,111	445		2000	2000	445

## Data Availability

The raw omics data for ROSMAP are available at https://adknowledgeportal.synapse.org/. The raw omics data for BRCA, LGG, and KIPAN are available at https://portal.gdc.cancer.gov/ (accessed on 29 December 2023). Codes are available at https://github.com/Yaolab-fantastic/TEMINET (accessed on 29 December 2023).

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
