# Peer review of "TEMINET: A Co-Informative and Trustworthy Multi-Omics Integration Network for Diagnostic Prediction"

_ijms, 2024, doi:10.3390/ijms25031655_

Round 1

Reviewer 1 Report

Comments and Suggestions for Authors

File attached

Reviewer 2 Report

Comments and Suggestions for Authors

This work introduces TEMINET, an innovative approach for the integration of multi-omics data, addressing the complexities arising from the expanding variety of omics information. This approach employs an intra-omics co-informative representation method coupled with a robust learning strategy to tackle the challenges associated with both intra- and inter-omics integration. Leveraging graph attention networks and a multi-level framework, it captures collective informativeness beyond pairwise relations, offering a comprehensive view of the omics landscape. The manuscript showcases TEMINET's effectiveness through experiments across four different diseases, using mRNA, methylation, and miRNA data, demonstrating its performance and robustness in classification tasks.

1) I think the authors should discuss and made possible comparisons with transformer models, e.g., PMID: 37798249; PMID: 37699885, which helps convince the superior results of TEMINET.

2) The method part is lack of details. More detailed descriptions are needed to explain the method.

3) I would recommend the authors to make figures with high resolution, as well as consistent font size throughout the manuscript.

Comments on the Quality of English Language

The overall writing has some formatting issues, like wording, spacing, and some redundancy. I suggest the authors check the grammar and avoid any typos. More importantly, the writing needs improvement for readers to understand more easily.

Reviewer 3 Report

Comments and Suggestions for Authors

Here, the authors discuss a novel multi-omics integration approach called TEMINET, which aims to improve diagnostic prediction for complex diseases. They describe the methodology behind TEMINET, which uses graph attention networks to capture co-informative intra-omics representations, and a trustworthy learning strategy to fuse inter-omics information based on uncertainty. They also report the results of applying TEMINET to four public benchmark datasets of different diseases, and compares its performance with existing methods. The article also identifies important biomarkers and analyzes their biological relevance. Finally, the authors concludes that TEMINET achieves superior performance and robustness in multi-omics prediction tasks, and demonstrates the potential of graph-based feature representation and uncertainty-based trustworthy learning integration.

Major points:

The authors propose a novel and innovative method that leverages both intra- and inter-omics information for multi-omics integration, which is a challenging and important problem in biomedical research.

They show that the proposed method outperforms state-of-the-art models on four different diseases using three types of omics data, and provides evidence of its stability and interpretability.

The article addresses a relevant and timely topic, as multi-omics data integration has emerged as a rapidly expanding research field, aiming to provide a more comprehensive understanding of the underlying biological processes and molecular mechanisms involved in complex diseases. This article would be suitable for the readership of the journal, and is an important contribution to the field.

Minor points:

The authors do not discuss the limitations of the proposed method, such as the scalability, generalizability, or computational efficiency, which are important aspects to consider for practical applications. This should be discussed further.

The article does not provide enough details on how the datasets were preprocessed, how the hyperparameters were tuned, or how the statistical significance of the results was assessed. This should be developed further in the main text.

Comments on the Quality of English Language

minor typos
